# A Novel Dual PI3K/mTOR Inhibitor, XIN-10, for the Treatment of Cancer

**DOI:** 10.3390/ijms241914821

**Published:** 2023-10-01

**Authors:** Leixuan Luo, Xin Sun, Yang Yang, Lulu Xia, Shiyu Wang, Yuxing Fu, Yuxuan Zhu, Shan Xu, Wufu Zhu

**Affiliations:** Jiangxi Provincial Key Laboratory of Drug Design and Evaluation, School of Pharmacy, Jiangxi Science & Technology Normal University, 605 Fenglin Road, Nanchang 330013, China; luoleixuan8@163.com (L.L.); sunxin_haha@163.com (X.S.); yangyang9879876@163.com (Y.Y.); 17839837110@163.com (L.X.); wangsy030101@163.com (S.W.); fformd@163.com (Y.F.); zyuxuan1998@163.com (Y.Z.)

**Keywords:** PI3Kα/mTOR dual inhibitor, cancer treatment, antitumor activity

## Abstract

An imbalance in PI3K/AKT/mTOR pathway signaling in humans often leads to cancer. Therefore, the investigation of anti-cancer medications that inhibit PI3K and mTOR has emerged as a significant area of research. The aim of this study was to explore the effect of XIN-10, a dual PI3K/mTOR inhibitor, on the growth as well as antiproliferation of tumor cells and to investigate the anti-tumor mechanism of XIN-10 by further exploration. We screened three cell lines for more in-depth exploration by MTT experiments. From the AO staining, cell cycle and apoptosis, we found that XIN-10 had a more obvious inhibitory effect on the MCF-7 breast cancer cell line and used this as a selection for more in-depth experiments. A series of in vitro and in vivo experiments showed that XIN-10 has superior antiproliferative activity compared with the positive drug GDC-0941. Meanwhile, through the results of protein blotting and PCR experiments, we concluded that XIN-10 can block the activation of the downstream pathway of mTOR by inhibiting the phosphorylation of AKT(S473) as well as having significant inhibitory effects on the gene exons of PI3K and mTOR. These results indicate that XIN-10 is a highly potent inhibitor with low toxicity and has a strong potential to be developed as a novel PI3Kα/mTOR dual inhibitor candidate for the treatment of positive breast cancer.

## 1. Introduction

Cancer begins with cell mutation, where infinitely proliferating cancer cells invade and destroy normal tissues and organs, affecting human physiological functions and even being fatal [1,2,3]. PI3K/AKT/mTOR is a major pathway in human cancer and is crucial in cell proliferation, metastasis, and metabolism, and blocking the related targets of this pathway can exert an inhibitory effect on cancer [4,5]. After the receptor on the cell membrane surface emits a signal, the p85 regulatory subunit of PI3K (phosphatidylinositol 3-kinase) is recruited to the cell plasma membrane, where it binds to the p110 catalytic subunit, which specifically catalyzes the conversion of PIP2 to PIP3 [6,7,8,9]. PIP3 binds to the PH structural domain of AKT (protein kinase B, PKB) [10,11]. AKT indirectly promotes the activation of mTOR (mammalian targets of rapamycin), which promotes cell growth and cell cycle progression [12,13]. Therefore, the development of drugs targeting key kinases of PI3K and mTOR has become a trend [14,15,16].

GDC-0941 [17,18,19] (Pictilisib), developed by Genentech, exhibits moderate selectivity with an IC50 of 3nM for both PI3Kα/δ. GDC-0941 induces autophagy and apoptosis in the majority of cancer cells. In in vivo experiments, its bioavailability can reach 78% after oral administration, with a high plasma protein binding rate. Unfortunately, GDC-0941 failed to achieve the primary goal of phase II clinical studies in breast cancer and led to serious adverse effects. Therefore, the subsequent development of GDC-0941 was terminated [20].

In the present study, we conducted a series of studies using XIN-10, which was derived from the modification of GDC-0941. We investigated its kinase inhibitory activity, cytotoxicity, and antitumor ability in nude mice. Additionally, in order to study the anti-tumor mechanism of XIN-10, we conducted a series of experiments including cell cycle analysis, apoptosis assays, PCR, and Western blotting.

## 2. Results

### 2.1. Evaluation of the Cytotoxicity 

To evaluate the in vitro antitumor activity of XIN-10, GDC-0941 was used as a positive control, 10 different human cancer cells (MCF-7, A549, NCI-H460, H1975, H2228, Hela, Hela-MDR, ovcar-3, U87MG, and MDA-MB-231) were selected as the tested cancer cells. As shown in Table 1 and Figure 1, the IC50 values of XIN-10 against these 10 tested cancer cell lines ranged from 1.702 ± 0.06 μM to 0.0582 ± 0.002 μM. We found that XIN-10 showed good activity against a variety of cancer cell lines. It was also observed that XIN-10 was more effective than the positive control drug GDC-0941 against almost all the tested cancer cell lines.

### 2.2. Evaluation of the Kinase Inhibitory Activities

We tested the activity of compound XIN-10 against a series of tyrosine kinases (VEGFR-2, c-Met, AKT1, and EGFRT790M/L858R) to assess its selectivity for PI3Kα and mTOR kinases, and the results are shown in Table 2 and in Figure 2: at a dose of 1 µM, compound XIN-10 inhibited PI3Kα kinase by up to 1 µM, and compound XIN-10 inhibited PI3Kα kinase by up to 92.9 ± 2.4% and mTOR kinase by 85.3 ± 2.9%. Moreover, the inhibitory effect of XIN-10 on VEGFR-2, c-Met, AKT1, and EGFRT790M/L858R kinases was weak, as shown in Table 3 the selectivity of XIN-10 for PI3K, mTOR and other tested kinases was more than 13.3-fold. At the same time, the kinase IC50 test for PI3Kα and mTOR was also performed for XIN-10 and the positive drug GDC-0941, and the kinase IC50 values of XIN-10 for PI3Kα and mTOR reached 0.0508 μM and 0.0214 μM as shown in Table 4, which were much higher than those of GDC-0941.

### 2.3. AO Staining

We have selected three cell lines for a more in-depth study of XIN-10 based on the results of our cytotoxicity assay. 

Figure 3 demonstrates the AO staining assay performed on MCF-7, Hela, and A549 cells exposed to XIN-10, indicating that XIN-10 has considerable ability to induce apoptosis in tumor cells. Observed under the microscope, the control group showed a consistent distribution of green fluorescence. Normal cells were structurally intact, with clear edges and tight distribution. After 6 h of exposure to 0.6 μM of XIN-10, A549 cells showed enhanced punctate fluorescence, indicating the generation of apoptotic vesicles. At a concentration of 1.2 μM, the gap between cells was significantly enlarged, while the cell fluorescence was significantly weakened, and the number of cells decreased from 149 to 81. Most of the cells were in the late stage of apoptosis or death. Similarly, in Hela cells, with the increase in compound XIN-10, the fluorescence gradually decreased, and the number of cells also decreased from 89 to 64. MCF-7 cells with different concentrations of XIN-10 had a similar apoptotic effect, with a decrease from 104 to 47 cells. Thus, all the findings suggest that XIN-10 is effective in triggering programmed cell death in A549 cells, Hela cells, and MCF-7 cells, and that this effect varies with dose.

### 2.4. Analysis of Cell Cycle

To guarantee the precision of the cell cycle inhibition caused by XIN-10, flow cytometry was employed to examine the cell cycle of MCF-7, A549, and Hela cell lines after treatment with varying concentrations of XIN-10. Figure 4 displays the outcomes of the cell cycle analysis for the three cancer cell lines that were treated with XIN-10, along with the distribution of each stage. It is evident from Figure 4 that as the concentration of XIN-10 increases, the percentage of S phase arrest in MCF-7 cells rises from 18.00% in the control group to 28.47% (0.3 μM), 35.41% (0.6 μM), and 39.39% (1.2 μM) in the treatment groups. This indicates a significant and dose-dependent impact of XIN-10 on S phase arrest in MCF-7 cells. Contrary to MCF-7 cells, the G0/G1 phase cell arrest rate of A549 cells escalated as the concentration of XIN-10 increased, rising from 58.30% in the control group to 70.81% (0.3 μM), 87.94% (0.6 μM), and 93.64% (1.2 μM). This suggests that XIN-10 can hinder the G0/G1 phase of A549 cells in a dose-dependent manner, which differs from its blocking mechanism in MCF-7 cells. Similar to A549 cancer cells, the cell arrest rate of Hela cells in the G0/G1 phase also increased with the augmentation of XIN-10 concentration, going from 69.62% in the control group to 78.24% (0.6 μM), 83.11% (0.6 μM), and 92.42% (1.2 μM). This indicates a dose-dependent blocking effect of XIN-10 on the G0/G1 phase of Hela cells.

Based on the aforementioned findings, it can be observed that XIN-10 has the ability to hinder the G0/G1 phase of A549 and Hela cells in a manner that is influenced by the concentration. Additionally, it can impede the S phase of MCF-7 cells in a way that is influenced by the dosage.

### 2.5. Apoptosis Inducing Activity Assay

We used FITC/PI double staining and flow cytometry to investigate the induction of apoptosis by compound XIN-10 in three cancer cell lines, MCF-7, A549, and Hela. After culturing cells for 24 h, cells were treated with the corresponding dose of compound for 48 h and analyzed by flow cytometry. The results of the treatment are shown in Figure 5 below.

The total apoptosis rate of the MCF-7 control group was 3.98%. Compound XIN-10 induced MCF-7 cell apoptosis at 0.2, 0.4, and 0.8 μM concentrations, with rates of 16.09%, 20.28%, and 31.61%, respectively. The rate of late-stage apoptosis increased by 15.52%. These results indicate that compound XIN-10 induces apoptosis in MCF-7 cells and mainly induces late-stage apoptosis. The total apoptosis rate of the A549 control group was 4.97%. Compound XIN-10 induced A549 cell apoptosis in a dose-dependent manner at concentrations of 0.2, 0.4, and 0.8 μM, with rates of 21.26%, 24.04%, and 26.79%, respectively. The rate of late-stage apoptosis increased by 4.34%. The apoptosis results indicate that compound XIN-10 induces late-stage apoptosis in A549 cells in a dose-dependent manner. The total apoptosis rate of the Hela control group was 4.56%. Compound XIN-10 induced Hela cell apoptosis in a dose-dependent manner at concentrations of 0.2, 0.4, and 0.8 μM, with rates of 16.19%, 19.37%, and 21.49%, respectively. The rate of late-stage apoptosis increased by 3.88%. The apoptosis results indicate that compound XIN-10 induces apoptosis in Hela cells in a dose-dependent manner and mainly induces late-stage apoptosis.

These results suggest that compound XIN-10 has a more significant apoptosis-inducing effect on MCF-7 cells, followed by A549 and Hela cells, mainly inducing late-stage apoptosis in cancer cells.

### 2.6. Time-Dependent Colony Experiments

Through AO staining and cell cycle and apoptosis experiments, we found that XIN-10 showed better effects on the MCF-7 cell line, so we decided to use the MCF-7 cell line for more in-depth pharmacological studies. To investigate the relationship between the extent of anticancer cell proliferation of the optimized compound XIN-10 and the duration of its action, a time-dependent experiment was performed on XIN-10 using the MTT method. MCF-7 cells were treated with the drug for 24 h, 48 h, and 72 h, respectively, and then MTT solution was added to measure the absorbance values. Three parallel experiments were performed, and the corresponding inhibition rates were calculated. As shown in Figure 6A, the anti-proliferative effect of XIN-10 on MCF-7 cells increased with increasing concentration and prolonged time of action. For XIN-10 at a concentration of 3.70 μg/mL, the inhibition rate reached 56.05% after 48 h of treatment, which was close to the inhibition rate at 72 h (63.97%). The highest inhibition rate reached 96.88% (72 h, 100 μg/mL).

The impact of compound XIN-10 on the colony of MCF-7 cells was tested through a colony experiment. The results in Figure 6B show that XIN-10 inhibits the formation of MCF-7 cell clones in a concentration-dependent manner. At a concentration of 1/4 of the IC50, XIN-10 can already inhibit about half of the cell clones, and at a concentration of 580nM, the formation of MCF-7 cell clones can be completely inhibited, indicating that XIN-10-treated MCF-7 cells have lost their ability to proliferate.

### 2.7. Quantitative Real-Time PCR

To further explore the mechanism of compound XIN-10 in inhibiting cancer cell proliferation, we treated MCF-7 cells with XIN-10 at concentrations of 0.3 and 0.6 μM for 24 h and analyzed the expression patterns of the PIK3CA and mTOR1 *genes* using fluorescence quantitative PCR. The results shown in Figure 7 indicate that XIN-10 at concentrations of 0.3 and 0.6 μM dose-dependently inhibited the expression of the PIK3CA and mTOR1 *genes*, with stronger effects than the control drug GDC-0941. Moreover, XIN-10 exhibited stronger inhibition of the mTOR1 gene than the PIK3CA gene, which is consistent with the results of kinase inhibition. These findings suggest that the ability of compound XIN-10 to regulate tumor cell proliferation is related to its inhibition of the expression of the PIK3CA and mTOR1 *genes*.

### 2.8. Cell Migration

The migration and repair ability of MCF-7 cells was examined using the scratch assay to study the effect of compound XIN-10. XIN-10 at concentrations of 0.3 μM and 0.6 μM, was contacted with cells for 24 h, and then photographs were taken.

As shown in Figure 8, after 24 h of cell culture, the cells in the control group moved towards the center region. Compared with the blank group, the cells in the 0.3 μM XIN-10-treated group migrated to the center at a shorter distance, and the cell migration rate was reduced from 62.6% to 42.4%. In contrast, the cell migration rate was reduced from 62.6% to 27.8% in the blank group compared with the 0.6 μM XIN-10-treated group. The results showed that XIN-10 had an inhibitory effect on cell migration and increasing the dose of XIN-10 enhanced its inhibitory effect on the migration and repair ability of MCF-7 cells.

### 2.9. JC-1

Maintaining the normal physiological function of mitochondria and cells requires a regular mitochondrial membrane potential (MMP). Hence, the reduction in MMP serves as a crucial indicator of cellular apoptosis. By detecting the transition from red fluorescence to green fluorescence of the JC-1 fluorescent probe from normal potential, it indicates the decrease in cell membrane potential and can detect cell apoptosis. We used JC-1 to detect the effects of compounds and XIN-10 on MMP of MCF-7 cells. As shown in Figure 9, compound XIN-10 can concentration-dependently reduce the red fluorescence of MCF-7 cells and convert it to green fluorescence.

### 2.10. ROS

Figure 10 demonstrates that the blank group’s MCF-7 cells exhibited minimal fluorescence signal, whereas MCF-7 cells treated with a low dosage of XIN-10 (0.2 μM) displayed a slight green fluorescence, indicating a limited generation of ROS. Following the administration of a potent dose of XIN-10 (0.4 μM), the cellular green fluorescence intensity exhibited a substantial augmentation, surpassing the impact of the reference medication at an equivalent concentration. The level of ROS in MCF-7 cells was increased in a dose-dependent manner by XIN-10, leading to apoptosis and successfully inhibiting tumor growth.

### 2.11. Western Blotting

Western blotting was used to determine the expression of phosphorylated AKT (p-AKT, S473) and phosphorylated mTOR (p-mTOR) at the S473 site in MCF-7 cells treated with XIN-10 to investigate its inhibition of the PI3K/AKT/mTOR signaling pathway in MCF-7 cells. As show in Figure 11, the results showed that the expression of p-AKT (S473) and downstream p-mTOR in MCF-7 cells treated with XIN-10 were dose-dependently inhibited. Overall, XIN-10 effectively inhibited the expression of p-AKT (S473) at a concentration of 0.1 μM, confirming the inhibitory effect of XIN-10 on the PI3K/AKT/mTOR pathway.

### 2.12. Molecular Docking

As shown in Figure 12A,B XIN-10 binds to the protein cavity of PI3Kα. Similarly, the morpholine oxygen forms an important hydrogen bond with the residue of Val851 in the hinge region (1.8 Å); the oxygen atom on the urea group in the affinity pocket has a hydrogen bond interaction with Lys802 (1.7 Å), the nitrogen atom forms a 2nd hydrogen bond interaction with ASP810 (2.08 Å), and the morpholine in the solvent region similarly has a hydrogen bond interaction with the residue of threonine THR-856 (2.1 Å). As shown in Figure 12C,D, XIN-10 binds to the mTOR protein. The morpholine oxygen forms a hydrogen bond with Val2240; the oxygen atom of the urea group has hydrogen bonding with ASP2357 and the nitrogen atom with ASP2195.

### 2.13. Hemolytic Toxicity Assay 

Initially, in order to evaluate the compatibility with blood, a test was conducted to measure the activity of hemolysis. In the experiments, positive and negative controls were employed, with 1% Triton X-100 and saline (NS) being used, respectively. Figure 13A demonstrates that XIN-10 displayed minimal hemolytic toxicity (less than 5% hemolysis) towards 2% of sheep blood erythrocytes following a 3 h incubation at concentrations of 16, 32, 64, 128, and 256 μg/mL. Overall, XIN-10 met the hemolytic toxicity requirements for intravenously administered compounds. Specifically, XIN-10 demonstrated less than 5% hemolysis at the maximum dose of 256 μg/mL, indicating minimal hemolytic toxicity and a favorable safety record. The findings from this study offered crucial direction for the subsequent phase of establishing the dosage concentration and dosage schedule of XIN-10.

### 2.14. Acute Oral Toxicity

Since XIN-10 showed excellent antiproliferative activity and tumor cell inhibition ability in vitro, we conducted further in vivo experimental studies on XIN-10. To evaluate the acute toxicity and safety of XIN-10 on normal mouse tissues, we conducted in vivo toxicity experiments on Kunming (KM) mice. Normal Kunming mice were divided into three groups: the drug administration group (XIN-10), the positive group (GDC-0941), and the blank group (NS), and three KM mice were set up in each group. After 7 days of intragastric administration (XIN-10), blood samples were collected from the eyes of KM mice and analyzed using a biochemical analyzer. Liver and kidney functions were assessed in the administered groups of mice. Liver function was assessed by total protein (TP), urea, glucose (GLU), creatinine (CREA), aspartate aminotransferase (AST), alanine aminotransferase (ALT), alkaline phosphatase (ALP), lactate dehydrogenase (LDH), and creatine kinase (CK) levels. As shown in Figure 13B–D, the levels of these parameters in the XIN-10 treated mice were not significantly different from those in the positive and blank groups. The biochemical parameters of the XIN-10 injection group were not significantly different from those of the control and positive control groups, indicating that all mice had normal liver and kidney functions.

Furthermore, in order to evaluate the toxicity of XIN-10 to the major organs of KM mice, H&E staining was performed on them and isolated major organs. The H&E staining results of the major organs of the administered group (XIN-10), the positive group (GDC-0941), and the blank group were compared. The H&E staining results in Figure 13E showed that the nuclei of cells in all groups were normal without any signs of cell damage or necrosis. The morphology and cell structure were clear, indicating that XIN-10 had no obvious toxic effects. These results were consistent with the assessment of liver and kidney functions. Therefore, the acute toxicity of XIN-10 in KM mice is extremely low, indicating that it has good biosafety and compatibility in vivo.

### 2.15. Inhibition of Growth of MCF-7 Cell Transplanted Tumors by Compound XIN-10 and Evaluation of Toxicity

We performed xenograft tumor nude mouse model experiments to more convincingly demonstrate the anti-tumor performance of the compounds in vivo. The transplanted tumors in each experimental group were isolated and placed neatly as shown in Figure 14A. There is no visible lesion in the transplanted tumor tissue. Overall, the individual size of transplanted tumors in the treatment group showed a decreasing trend compared to the blank group.

The length and diameter of the transplanted tumor in nude mice were recorded every 2 days, and the calculated volume was plotted into a tumor growth curve. As shown in Figure 14E, in the MCF-7 cell tumor bearing nude mouse model, compared to the blank control group, the transplanted tumor volume in the XIN-10 treatment group and the positive control drug GDC-0941 treatment group was significantly reduced. 

As shown in Figure 14B–C, the average tumor weight of the control group was 1.48 g; the average tumor weight of the positive control drug GDC-0941 treatment group (75 mg/kg) was 0.35 g, and the tumor inhibition rate reached 74.7%; the average tumor weight of the XIN-10 high-dose treatment group (75 mg/kg) was 0.61 g, and the tumor inhibition rate was 55.8%. There was a significant difference in the transplanted tumor volume between the positive control drug GDC-0941 treatment group (75 mg/kg) and the blank group, with a *p*-value of 0.0084; there was also a significant difference in the transplanted tumor volume between the XIN-10 treatment group (75 mg/kg) and the blank group, with a *p*-value of 0.0346. This indicates that compound XIN-10 can effectively inhibit the growth of MCF-7 cell transplantation tumors, but the inhibition rate is lower than GDC-0941.

In addition, we processed hematoxylin–eosin staining (H&E staining) by dehydrating sections of heart, liver, spleen, lung, and kidney tissues from each experimental group of nude mice to determine the pathomorphological effects of organ tissues. The staining results in Figure 15 show that no pathological reactions were seen in the organs of the XIN-10 treated and positive drug treated groups, and there was no significant difference in the staining of the treated groups compared with the blank control group, indicating that the compounds XIN-10 and GDC-0941 had no pathological damage and no significant toxic reactions on the organs of nude mice.

### 2.16. Hematoxylin–Eosin Staining of MCF-7 Transplanted Tumors in Nude Mice (HE Staining)

The stripped graft tumor tissues were processed by hematoxylin–eosin staining (H&E staining). As show in Figure 16, the staining results of each transplantation tumor showed that the tumor cells in the blank group were structurally intact, oval or shuttle-shaped, uniformly distributed and tightly arranged, and no necrotic abnormal reactions were observed. The XIN-10 treatment group (75 mg/kg) and GDC-0941 treatment group (75 mg/kg) showed increased cellular fibrous organization, larger intercellular spaces, reduced cell numbers, and different sizes, indicating that the compounds XIN-10 and GDC-0941 had a toxic response on MCF-7 tumor tissue, inducing nuclear sequestration and shrinking the cytoplasmic volume of cancer cells to block their proliferation.

### 2.17. Immunohistochemical Staining of MCF-7 Cell Transplanted Tumors

The rates of bax, caspase-3, bcl-2, ki67, and p53-positive cells were examined by immunohistochemistry on MCF-7 cell transplant tumor tissues from nude mice. Among them, the bax gene and caspase3 gene promoted the apoptosis of MCF-7 cells, the Bcl-2 gene had the effect of inhibiting the apoptosis of MCF-7 cells, the Ki67 gene could promote the proliferation of MCF-7 cells, and P53 was an oncogene. The immunohistochemical results are shown in Figure 17. The results showed that the staining area (brown part) of caspase-3 was increased in the XIN-10 treatment group and the positive administration group compared with the blank group. While the expression level of the bax gene was not significantly different between the experimental groups. For the bcl-2 gene, there was no significant difference between the experimental groups. ki67 gene results showed a significant decrease in the stained area in the XIN-10-treated group and the positive-treated group compared with the blank group. For the P53 gene, the XIN-10 treatment group significantly increased the rate of P53-positive cells in MCF-7 cell transplantation tumors, which was much higher than that in the positive treatment group. In summary, the images of caspase-3, ki67, and p53 can show that XIN-10 has a gene-level inhibitory effect on MCF-7 breast cancer cells, and the effect is stronger than that of the positive control GDC-0941.

### 2.18. PCR Assay of MCF-7 Cell Transplantation Tumor 

The effect of XIN-10 and GDC-0941 on the expression of PIK3CA(p110) and mTOR1 in transplanted tumors was examined by PCR experiments on MCF-7 transplanted tumor tissues from nude mice. As shown in Figure 18, the expression of PIK3CA and mTOR1 was significantly reduced in the XIN-10 treatment group, and there was a dose-dependent decrease in their expression. The expressions of PIK3CA and mTOR1 were significantly different in each experimental group compared with the blank group. It indicates that the inhibition of PIK3CA and mTOR1 by the compounds achieved anti-tumor effects. Overall, the inhibition of PIK3CA and mTOR1 expression in MCF-7 transplanted tumor tissues was stronger in the GDC-0941 treatment group than in the XIN-10 treatment group, which was consistent with the results of tumor suppression rate of the compounds.

## 3. Discussion

In summary, this study reports that the dual PI3K/mTOR inhibitor XIN-10 can effectively inhibit the growth of MCF-7 cells in in vitro and in vivo settings. We conducted in vitro and in vivo antitumor activity studies on the dual PI3K/MOTR inhibitor XIN-10. The toxicity of XIN-10 on nine cell lines, including MCF-7 and Hela, was studied by the MTT method. In-depth pharmacological experiments were performed on compound XIN-10 to explore its mechanism of action. AO staining and FITC/PI double staining apoptosis assays showed that XIN-10 could induce apoptosis in MCF-7 and other cells. The cycle assay showed that it could block the proliferation of MCF-7 cells in S phase and also block the proliferation of A549 and Hela cells in G0/G1 phase. Reactive oxygen assay and mitochondrial membrane potential assay showed that it could increase ROS level, decrease mitochondrial membrane potential, and induce apoptosis in MCF-7 cells. Colony and migration assays showed that XIN-10 could effectively inhibit the proliferation and migration of MCF-7 cells. Fluorescence quantitative PCR experiments showed that it could significantly inhibit the expression of PI3K and mTOR *genes* in MCF-7 cells. In addition, Western blotting results showed that it inhibited phosphorylation of AKT-S473 in a dose-dependent manner and blocked phosphorylation of the down-stream pathway mTOR. The antitumor activity of compound XIN-10 was investigated in nude mice using the MCF-7 cell transplantation tumor model. The mean tumor weight of the XIN-10 treated group (75 mg/kg) was significantly different from that of the blank group, and the tumor inhibition rate was 55.8%. No pathological reaction was found in the organs of nude mice by HE staining. TUNEL staining and immunohistochemical assay of transplanted tumors showed that XIN-10 induced apoptosis in MCF-7 cell transplanted tumors. PCR results showed that it significantly inhibited the expression of PI3K and mTOR in MCF-7 transplanted tumors. Taken together, compound XIN-10 is a promising therapeutic approach as a dual PI3K/mTOR inhibitor for the treatment of breast cancer.

The mTOR single inhibitors often trigger negative feedback regulation of AKT when inhibiting mTOR, thereby reactivating the PI3K pathway. Although PI3K selective inhibitors can avoid negative feedback regulation of AKT, they may be affected by the catalytic subunit p110 α mutations leading to drug resistance. In addition, PI3K monotherapy affects insulin metabolism, leading to side effects such as hyperglycemia, anorexia, and nausea. Furthermore, PI3K/mTOR (especially PI3K) α/mTOR) dual ATP competitive inhibitors can directly act on key target proteins PI3K and mTOR, thereby more efficiently inhibiting the PI3K/Akt/mTOR signaling pathway, effectively preventing other factors from activating the PI3K pathway, and reducing the resistance and side effects produced by single inhibitors [21,22,23]. There are currently relevant reports on PI3K/mTOR dual inhibitors including Gedatolisib, Omipalisib, Apitolisib, etc. [24,25,26], all of which are in Phase I or Phase II clinical stages. There are currently no small molecule inhibitors of PI3K/mTOR on the market. As a modified drug of the PI3K single inhibitor GDC-0941, XIN-10 has stronger MTT and qPCR effects than GDC-041, and its response to the mTOR target is also significantly higher than the latter. It has the same inhibitory effect as PI3K in terms of kinase inhibition activity; therefore, XIN-10 has strong potential as a dual inhibitor of PI3K/mTOR, and the development of dual inhibitors of PI3K/mTOR can serve as a reference for cancer and research.

## 4. Methods and Materials

### 4.1. Reagents and Installations

The synthesis of XIN-10 is described in the Appendix A. The samples were confirmed for their purity and identification through the use of high-performance liquid chromatography, mass spectrometry, as well as nuclear magnetic resonance 1H and 13C spectroscopy. High performance liquid chromatograph purchased from Agilent (Beijing, China). The tandem quadrupole (triple quadrupole) mass spectrometer was purchased from Waters (Shanghai, China). Nuclear magnetic resonance NMR spectrometer purchased from Bruker (Beijing, China). The compound GDC-0941 was acquired from Sigma Chemical (St. Louis, MO, USA). We bought RPMI 1640 medium, DMEM high sugar medium, PBS buffer, and the penicillin–streptomycin combination from Solarbio (Beijing, China) for cancer cell culture reagents. Biological Industries (Israel) was the source of the fetal bovine serum that was acquired. Gibco (Waltham, MA, USA) provided the Trypsin-EDTA, and Keygen Biotechnology (Nanjin, China) supplied the AO Acridine Orange staining assay kit, PI/RnaseA cell cycle assay kit, and Annexin V-FITC/PI apoptosis assay kit. Yuhuan Southern Reagent Co. (Yuhuan, China) supplied 4% erythrocytes from sheep blood, while crystal violet, 10% TritonX-100, Hoechst 33,342, and ROS kits were obtained from Bayutai, China. The MMP assay kit, the fixative made from paraformaldehyde, the kit for isolating cell mitochondria using JC-1, and the 0.25% trypsin (EDTA-free) were bought from Servicebio (Wuhan, China). Takara (Shiga, Japan) was the source of the PCR experiments, which involved the acquisition of the PrimeScript RT reagent kit, total RNA extraction kit, DEPC water, agarose, sequence-specific primers, and TB Green Premix Ex Taq kits. Antibodies for the Western blot experiments that were associated were bought from Affinity (Changzhou, China). Servicebio (Wuhan, China) provided rIPA lysate, PMSF, 5 × SDS-PAGE protein loading buffer, and SDS-PAGE gel preparation kits. Beyotime (Shanghai, China) was the source of the purchased goat anti-rabbit IgG, bovine V albumin, skim milk powder, ECL A solution, and ECL B solution. The mice and nude mice used in the animal experiments were purchased from Silaikejingda (Changsha, China). IX71 fluorescence inverted microscope purchased from OLYMPUS (Tokyo, Japan). SPECTRA MAX plus Enzyme Labeler purchased from Thermo Scientific (Waltham, MA, USA). C6 flow cytometer purchased from BD (Shanghai, China). DYY-6C electrophoresis analyzer purchased from Liuyi Biotechnology (Beijing, China). PCR instruments and agarose gel electrophoresis tank were purchased from Bio-Rad (Hercules, CA, USA). Fully automated gel imager purchased from SYNGENE (Cambridge, UK).

### 4.2. Synthesis of 1-(1-methylpyrazolin-4-yl)-3-(4-(4-morpholin-6-(5-(morpholinomethyl)thiophen-2-yl)-1,3,5-triazin-2-yl)phenyl)urea (XIN-10)

XIN-10 was produced through a series of chemical reactions involving morpholine substitution, Suzuki coupling, acylation, reductive amination with morpholine, and aminourea generation, starting from 2,4,6-trichloro-1,3,5-triazine.

A solid of a pale-yellow color was obtained with a yield of 67.2%. M.P.: 156.4–157.7 °C. In the 1H NMR spectrum (at 400 MHz, using DMSO-d6 as the solvent), the chemical shifts are as follows: 9.04 (singlet, 1H), 8.48 (singlet, 1H), 8.39 (doublet, *J* = 8.3 Hz, 2H), 8.01 (doublet, *J* = 3.5 Hz, 1H), 7.80 (singlet, 1H), 7.62 (doublet, *J* = 8.4 Hz, 2H), 7.41 (singlet, 1H), 7.11 (doublet, *J* = 3.4 Hz, 1H), 3.95 (doublet, *J* = 17.5 Hz, 4H), 3.81 (singlet, 3H), 3.75 (doublet, *J* = 6.6 Hz, 6H), 3.66–3.59 (multiplet, 4H), and 2.47 (doublet, *J* = 5.5 Hz, 4H). The 13C NMR spectrum (101 MHz, DMSO-d6) shows peaks at chemical shifts of 168.98, 165.88, 163.41, 151.35, 147.56, 143.22, 140.38, 129.78, 129.20, 128.76 (2, C), 127.99, 126.89, 120.90, 120.40, 116.59 (2, C), 65.58 (2, C), 65.36 (2, C), 56.47, 52.36 (2, C), 42.72 (2, C), and 38.06.

The calculated mass-to-charge ratio (m/z) for the TOF MS ES + (M + H) + of C22H27N5O2S is 562.2349, which was also found to be 562.2349. HPLC (CH3CN/H2O (KH2PO4 = 20 mM)): tR: 5.643 min, purity: 97.73%. Additional information, including specific characterization data and spectra, can be located in the Appendix A.

### 4.3. In Vitro Assays to Measure the Inhibitory Activity against Cell Proliferation

The anti-proliferative activity of XIN-10 on various cancer cell lines including MCF-7, ovcar, A549, NCI-H460, H1975, H2228, U87MG, Hela, Hela-mdr, and human normal hepatocellular carcinoma cells LO-2 was evaluated using the MTT assay. GDC-0941 was used as a positive control. Our previous research formed the basis for the specific procedure [27].

### 4.4. Assays for Inhibiting Enzymes

Sandia Medical Technology (Shanghai, China) Co. conducted inhibitory activity assays for PI3Kα, mTOR, Met, AKT1, and EGFR kinases. In short, various compounds and enzymes were diluted in kinase buffer to the specified concentrations.

### 4.5. AO Staining Method

Cells that were multiplying exponentially were transformed into a suspension of cells and then introduced into 24-well plates with a density of 2.0 × 104. The cells were introduced into 24-well plates with a concentration ranging from 2.0 × 104 to 4.0 × 104 cells per well. Following the specified duration of drug activity, the cells were incubated in the incubator for 15 min using acridine orange (AO) added to the culture solution, and cell morphology and cell fluorescence were observed under an inverted microscope. Acridine Orange is able to penetrate conventional cell membranes and can be used to stain DNA and RNA in the nucleus. Under the microscope, AO develops an intact, uniform green or yellow-green fluorescence. Apoptotic cells exhibit chromatin condensation or fragmentation, resulting in the formation of apoptotic vesicles with mixed-sized fragments. These vesicles are subsequently stained by AO with intense yellow-green fluorescence. In contrast, necrotic cells show reduced or complete loss of fluorescence after AO staining.

### 4.6. Cell Cycle Assay

The Cell Cycle Analysis Kit (Beyotime Biotechnology Shanghai, China) was used to identify the phase of the cell cycle. The three cell lines were treated with varying amounts of XIN-10 (0.3 and 0.6 μM) in this study, and subsequently examined using flow cytometry after 24 and 48 h of treatment. To summarize, cells in the logarithmic growth phase were introduced into 6-well plates at a density of 1 × 10^6^ cells per well. After 24 h, a specific amount of XIN-10 was introduced into the medium containing the cells. After 48 h of incubation, the cells that were in opposition were made into a suspension and then washed two times using a phosphate buffered solution (PBS). Cells were collected, fixed in 70% ice ethanol, and then incubated at 4 °C for 2 h or overnight. The cells were placed in an incubator and then rinsed two times with PBS. In a dimly lit setting, the prearranged dye (RNase A propidium iodide = 91) was introduced to the cells. In dark conditions, the response time exceeded 30–60 min. A flow cytometer (BD Accuri™C6, Piscataway, NJ, USA) was utilized to gather cells (1 × 10^4^). Analysis of the last phase of the cell cycle was conducted.

### 4.7. Cell Apoptosis

The Annexin V-FITC/PI Apoptosis Assay Kit (Beyotime Biotechnology) was utilized to conduct the apoptosis assay. Following the cyclic experimental procedure mentioned earlier, cells were seeded in 6-well plates and exposed to compound XIN-10 for a duration of 48 h. Subsequently, the cells were digested, centrifuged, rinsed twice with PBS, and the supernatants were discarded. Further operations were conducted in a dim setting. The cells were treated with Annexin V-FITC and PI staining solutions in a dark setting for 15 min, and flow cytometry (BD Accuri™C6) was primarily used to detect apoptosis.

### 4.8. Cell Colony Formation Assay

A total of 5000 cells were cultured in 6-well plates, where the cells were in logarithmic growth phase. Twenty-four hours later XIN-10 was added and incubated. After 48 h, the cells were substituted every 2 days and the condition of the cells was monitored for a period of 14 days, or until the majority of individual clones contained over 50 cells. Afterwards, the cells were rinsed twice with PBS and captured using a microscope to be counted. To fix the cells, 2 mL of 4% paraformaldehyde was added per well for 15–30 min, followed by aspiration of the paraformaldehyde. After aspirating paraformaldehyde, it was washed twice with PBS. Then, 1 mL of crystalline violet staining solution was added to each well, followed by a 15 min staining period and subsequent photography. After a duration of 15 min, photographs were taken of the complete six-well plate as well as each individual well. Images captured every individual well. Calculation was performed to determine the rate of cell clone formation among various groups. Mathematical analysis was used to determine the effect of the drug on the growth of cancer cells.

### 4.9. Real-Time PCR

The relative expression of PIK3CA (P110) and mTOR1 mRNA was detected using real-time PCR. TRIzol reagent (Life Technologies Shanghai, China) was used to extract total RNA from drug-spiked cells following the provided instructions. The cDNA was acquired through reverse transcription utilizing the PrimeScript RT kit and gDNA Eraser from Takara. To measure the amounts of transcripts, a PrimeScript RT Master Mix from Takara and a real-time PCR system called Stratagene Mx3000p were utilized for SYBR Green qPCR. In humans, the forward primer for PI3K is GGGCTTTCTGTCTCCTCTAAAC, while the reverse primer is ATGTCTGGGTTCTCCCAATTC. The human mTOR forward primer sequence is GCTGTGAGGTCTGAGTTTAAGG, while the reverse primer sequence is ATTGCCTTCTGCCTCTTATGG. To control the internal mRNA level, GAPDH was utilized. Fold change relative to the internal control is used to express the results.

### 4.10. Cell Scratch Assay

To investigate the impact of compound XIN-10 on the migratory and reparative capabilities of MCF-7 cells, the scratch assay was employed. The ability of MCF-7 cells to migrate and repair is being studied. On the backside of a 6-well plate, create parallel and uniform scratches, and each well was inoculated with around 5 × 10^5^ cells. After the cells were fully developed, the vertical tip of the gun was slid along the indicated path and the markings were traced with a fingernail. Rinse thoroughly twice with PBS to remove damaged cells. After incubating the cells, the medium was changed and the cells were observed under a microscope and images were taken. The cells were then observed under a microscope and photographed, and then the culture was continued in the incubator.The process of incubation was prolonged inside the incubator. After a period of 24 h, the cells were extracted, examined using a microscope, and captured in photographs.

### 4.11. JC-1 Assay

JC-1 was employed to assess the impact of compound XIN-10 on MMP in MCF-7 cell lines. Maintaining a normal physiological function of mitochondria and cells requires a standard mitochondrial membrane potential (MMP). A prerequisite for normal physiological function of mitochondria and cells. Hence, a reduction in MMP serves as a significant indicator of apoptosis. The cell suspensions were placed in 24-well plates with a cell density ranging from 2.0 × 10^4^ to 4.0 × 10^4^ cells per well. They were then incubated in an incubator for a duration of 24 h. After 24 h of incubation in the incubator, XIN-10 was added to the culture. After a period of 24 to 36 h, the cells were treated with the JC-1 kit and examined using a microscope. The JC-1 kit was examined using a fluorescent microscope (OLYMPUS, Tokyo, Japan).

### 4.12. ROS

The cells were taken in log proliferation phase, the cell suspension was inoculated at a density of 2.0 × 10^5^ cells/3 mL/well in a 6-well plate and incubated in the incubator for 24 h. The culture solution containing the compounds was added, and the solution was incubated in the incubator for the indicated time. The old culture medium was discarded, cells were washed with PBS 2 times and then stained with Hoechst 33,342 for 10 min. After washing the cells with PBS 2 times, 1 mL of culture medium mixed with DCFH-DA was added to each well. The paraformaldehyde was aspirated off, the cells were washed with PBS 2 times, and then cells were stained with crystal violet staining for 1 mL/well, and left to stand for 15 min. After washing the cells with PBS 2 times, the cells were observed under the fluorescence microscope, and pictures were taken.

### 4.13. Western Blot Method

The expression of PI3K, AKT, mTOR, S6K1, and other related proteins was detected using Western blotting. Proteins such as mTOR and S6K1, along with other proteins that are associated with them, were obtained from tumor tissue cells, which were broken down in a lysis solution for 30 min. After that, the mixture was centrifuged at 4 °C and 12,000 r/min for 5 min. Following centrifugation, the liquid portion above the sediment was collected. Following the determination of protein concentration using the BCA technique, the samples underwent gel electrophoresis followed by transferring the membrane. A closure containing 5% skim milk powder was used. The primary antibody was applied at a dilution of 1:1 and incubated overnight. The primary antibody was also applied at a dilution of 1:1000. After that, the secondary antibody was applied at a dilution of 1:10,000 and incubated for 2 h at room temperature. The gel was then developed using a chemiluminescence-based gel imaging system. The bands were quantified using Quantity One v4.6.6 and CellSens (version 1.6.0) Version 4.6.6, Image J 180, and various other software options are available for quantification.

### 4.14. Molecular Modeling

The protein ligands PI3K (PI3K PDB: 4L23 and mTOR PDB: 4JT6) were downloaded from the RCSB database (http//www.wwpdb.org/, accessed on 24 August 2022). The PDB was processed with Autodock 4.2 software. Preparation of the molecule: the small molecule structure was drawn with the software ChemDraw 3D (Version 16.0.0.82). It was saved in PDB format, and the structure was optimised with the software Autodock 4.2 to obtain a lower energy molecular conformation. Docking: the ligand and the molecule were positioned, energy optimised and docked. The docking results with good conformation were screened and saved. Modification by Pymol software(Version 2.0).

### 4.15. Assessment of Hemolytic Toxicity

Testing the safety of an injection involves observing the occurrence of a hemolytic reaction. To remove the fluid used to preserve the red blood cells, a centrifuge tube containing 1.5 mL of sheep blood erythrocytes with a concentration of 2% was spun at 3000 rpm for 5 min. Next, a 0.9% saline solution is added to the centrifuge tube to disperse the red blood cells. To serve as a negative control, physiological saline was utilized, while 1% TritonX-100 was employed as a positive control. The corresponding concentration of compound XIN-10 was added to 2% sheep blood erythrocyte saline. The mixture mentioned above was placed in an incubator for one hour at a temperature of 37 degrees Celsius. During this time, it was monitored for any signs of hemolysis or agglutination reactions. Under chilled conditions, the supernatant was spun at 3000 rpm for 5 min, and 500 μL of the resulting liquid was collected. Subsequently, the absorbance at 540 nm was determined using a UV spectrophotometer. In the end, the rate of hemolysis was determined using the provided formula. The percentage of hemolysis rate is calculated by subtracting the negative control group from the experimental group and dividing it by the difference between the positive control group and the negative control group.

### 4.16. Mouse Toxicity Experiments

Three mice from Kunming were divided into three groups: the blank group (treated with 0.9% saline), the positive group (treated with GDC-0941), and the XIN-10 group. The dose administered to each group was 75 mg/kg. Following 8 days of gavage treatment (administered 4 times every second day), blood samples were obtained from the mice’s ocular region. Serum was obtained by centrifugation after collecting blood from the eyes of mice. A preoperative, 10-item, quantitative assay kit and a biochemical analyzer were utilized to evaluate the serum, liver, and kidney functions. Histological analysis of mice involved dissection of the animals followed by staining their organs with hematoxylin–eosin acid (H&E staining).

### 4.17. Tumor Xenograft Experiments

All animal research conducted adhered to the ARRIVE guidelines and received approval from the animal ethics committee. Cell inoculation was used to construct nude mice tumor models using 18 BALB/c nude mice (female, 4–6 weeks) obtained from Hunan Slaughter and Jingda Laboratory Animal Co., all of which were thymus-free. Cells from inoculated nude mice were selected at logarithmic growth stages. Following the process of digestion, centrifugation, and cell counting, the cells that were centrifuged were ultimately suspended in saline solution. Subcutaneously injecting 0.2 mL of the cell suspension (5 × 10^6^ cells) into the right ventral region of the axilla was performed in the mice. The medication was administered orally for treatment. The length and width of the tumor were measured every two days and the volume was calculated using the ellipsoidal volume formula. Upon reaching a tumor volume of 150–180 mm^3^, the mice were administered compounds XIN-10 (75 mg/kg) and GDC-0941 (75 mg/kg) orally for treatment. Medication was given by mouth every second day for a period of 3 weeks. Afterwards, the mice underwent dissection. Wuhan Xavier conducted H&E staining and immunohistochemistry experiments on their tumor tissues and organ sections. 

### 4.18. Statistical Analysis

The IC50 values of the compounds were determined by first analyzing the MTT absorbance data to obtain the inhibition rate and then establishing a probit model by normalizing the data with IBM SPSS Statistics 22. Experiments such as cell migration and AO staining were carried out by using imageJ(Version java6 20170530ver) for the recording of the area and the number of cells. The data were plotted by using Graphpad Prism 8.0.2 software and significant difference analysis was conducted to obtain the *p*-value by its t-test analysis. The Western blots were quantified using Quantity One v4.6.6.

## Figures and Tables

**Figure 1 ijms-24-14821-f001:**
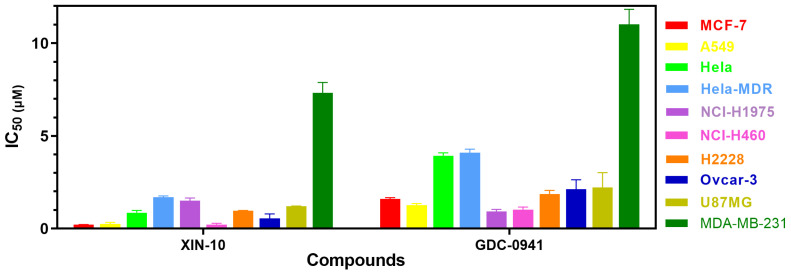
The cytotoxic effects of XIN-10 on 10 cancer cell lines were evaluated using MTT assay for 72 h, measuring the IC50 in micromolar concentration.

**Figure 2 ijms-24-14821-f002:**
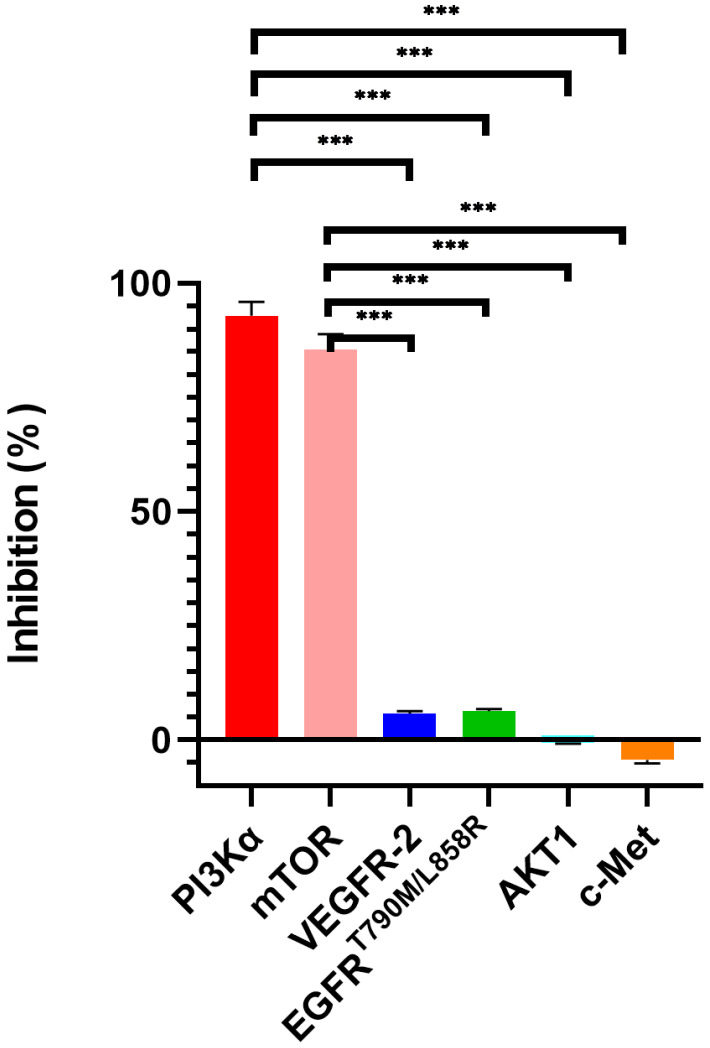
Tyrosine kinase selectivity of XIN-10. *** *p* < 0.001 indicates a very significant difference for PI3K α and mTOR compared to the other four tested kinases.

**Figure 3 ijms-24-14821-f003:**
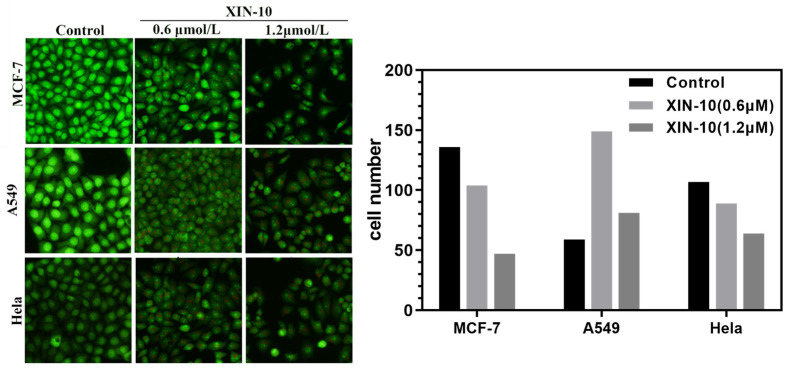
The AO apoptosis test (200×) and cell count of compound XIN-10 on MCF-7, A549, and Hela cells showed that XIN-10 promoted cell apoptosis with increasing concentration.

**Figure 4 ijms-24-14821-f004:**
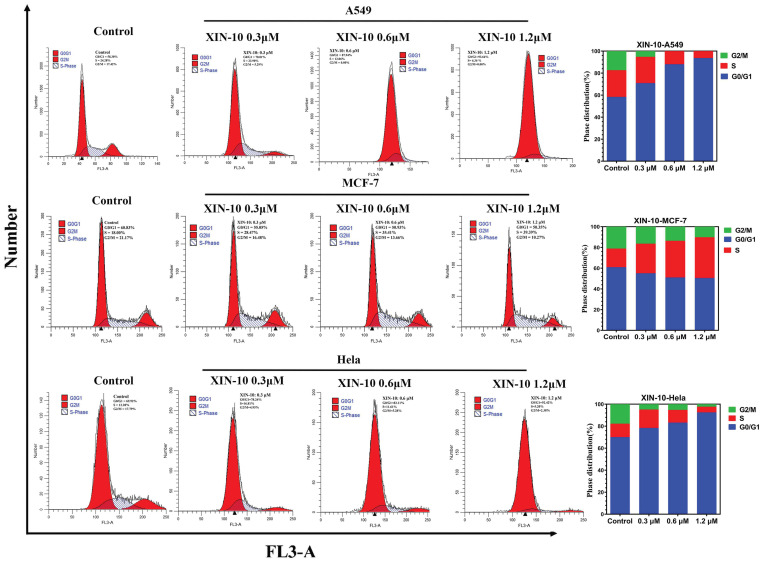
Graphs showing effect of compounds XIN-10 (with concentration of 0, 0.3, 0.6, and 1.2 µM) on A549, MCF-7, and Hela cells cycle distribution after 48 h treatment.

**Figure 5 ijms-24-14821-f005:**
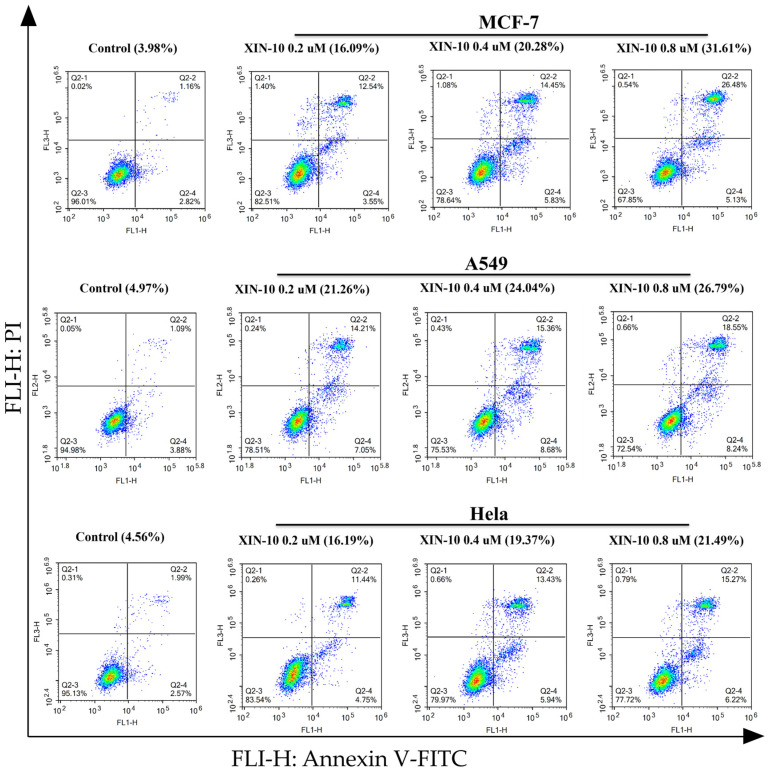
The cell apoptosis was quantitatively analyzed using Annexin V -FITC/PI double staining of compounds XIN-10 at concentrations of 0, 0.2, 0.4, and 0.8 µM in MCF-7, A549, and Hela cells.

**Figure 6 ijms-24-14821-f006:**
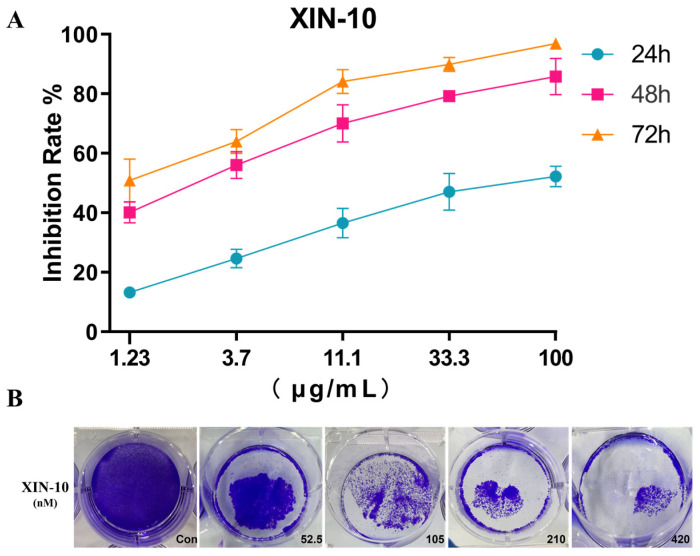
(**A**) The time-dependent experiment of compound XIN-10 on MCF-7 cells showed that the inhibitory activity increased with increasing concentration and prolonged action time. (**B**) Colony formation experiments showed that XIN-10 inhibits the formation of MCF-7 cell clones in a concentration-dependent manner.

**Figure 7 ijms-24-14821-f007:**
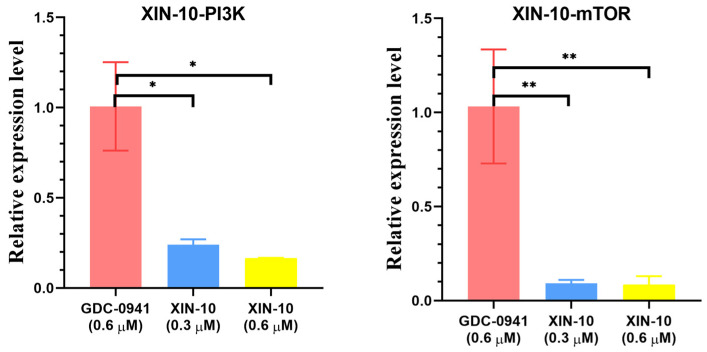
Analysis of the expression patterns of XIN-10 in MCF-7 cells for PI3K and mTOR. * *p* < 0.05 indicates a significant difference in PCR expression level between XIN10 and GDC-041. ** *p* < 0.01 indicates that XIN10 has a highly significant difference in relative expression level for GDC-041.

**Figure 8 ijms-24-14821-f008:**
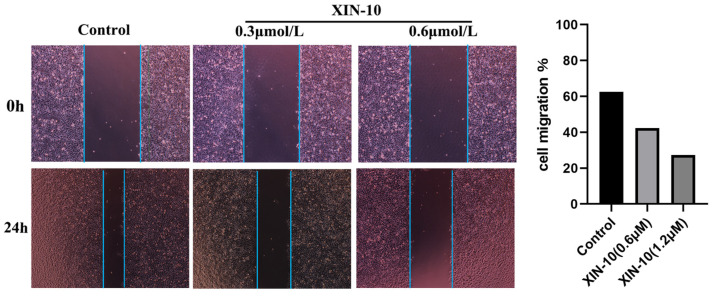
Compound XIN-10 exhibits concentration-dependent enhanced migration inhibition on the migration image and migration rate of MCF-7 cells (40×).

**Figure 9 ijms-24-14821-f009:**
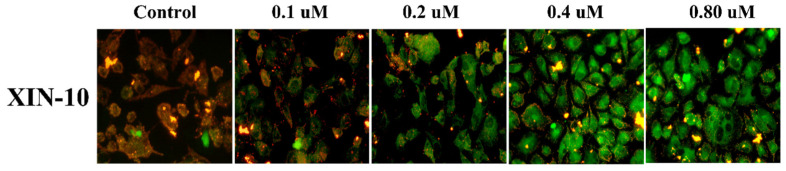
Effect of compound XIN-10 on MMP of MCF-7 cells (400×).

**Figure 10 ijms-24-14821-f010:**
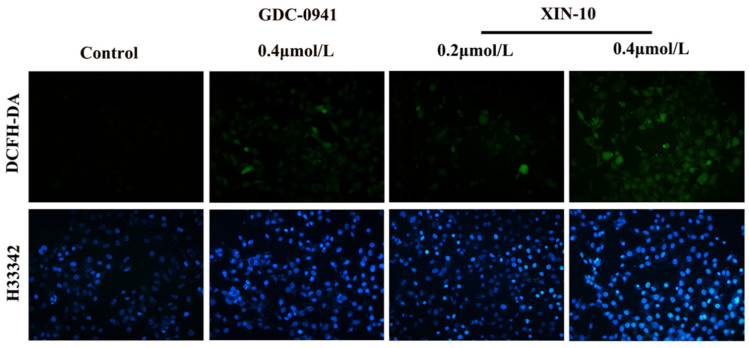
Effect of compound XIN-10 on reactive oxygen species in MCF-7 cells (100×).

**Figure 11 ijms-24-14821-f011:**
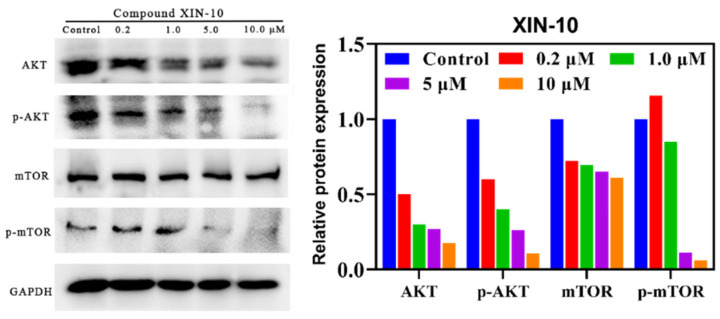
After 24 h of exposure, the impact of compound XIN-10 on the phosphorylation level of PI3K and mTOR in MCF-7 cells.

**Figure 12 ijms-24-14821-f012:**
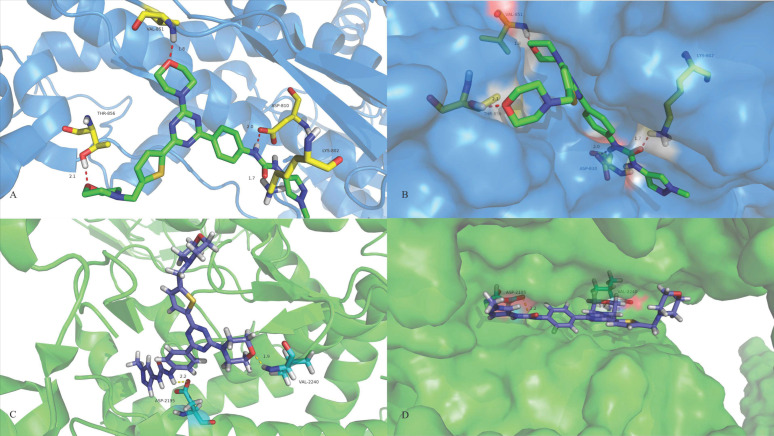
The docking mode for compound XIN-10. (**A**,**B**) The PI3Kα (PDB code 4L23) is bound by the XIN-10 in a binding model. (**C**,**D**) The mTOR binding model of an XIN-10 representative bound to mTOR (PDB code 4JT6).

**Figure 13 ijms-24-14821-f013:**
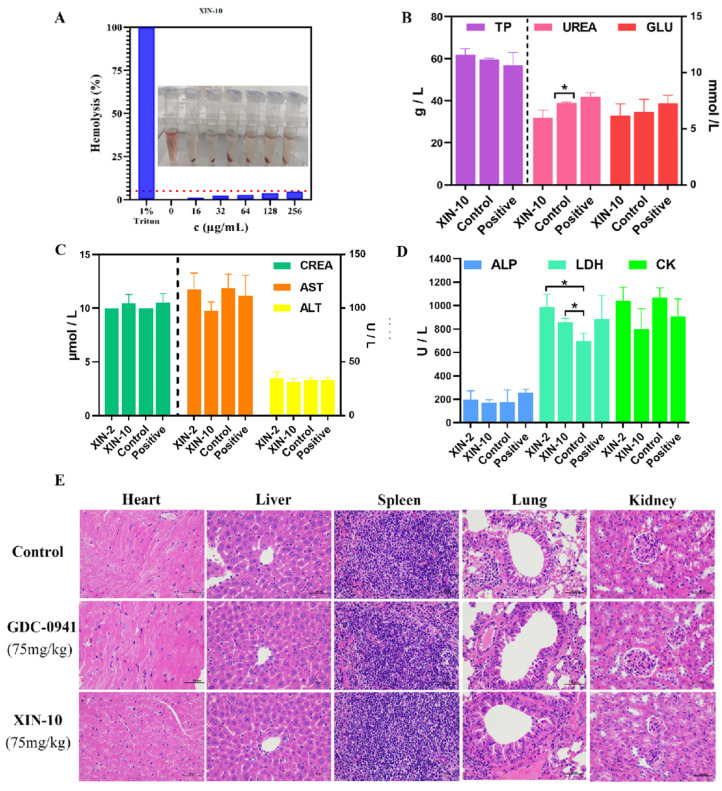
(**A**) Hemolytic toxicity study of XIN-10. (**B**–**D**) Blood biochemical analysis results of mice in the blank group (0.9% saline), XIN-10 (75 mg/kg), and positive administration group (GDC-0941, 75 mg/kg). * *p* < 0.05 indicates a significant difference between the treatment group and the blank control group (**E**) Hematoxylin–eosin staining (H&E staining) of major organs of different groups of mice, including heart, liver, spleen, lung, and kidney (Scale bar, 50 µm).

**Figure 14 ijms-24-14821-f014:**
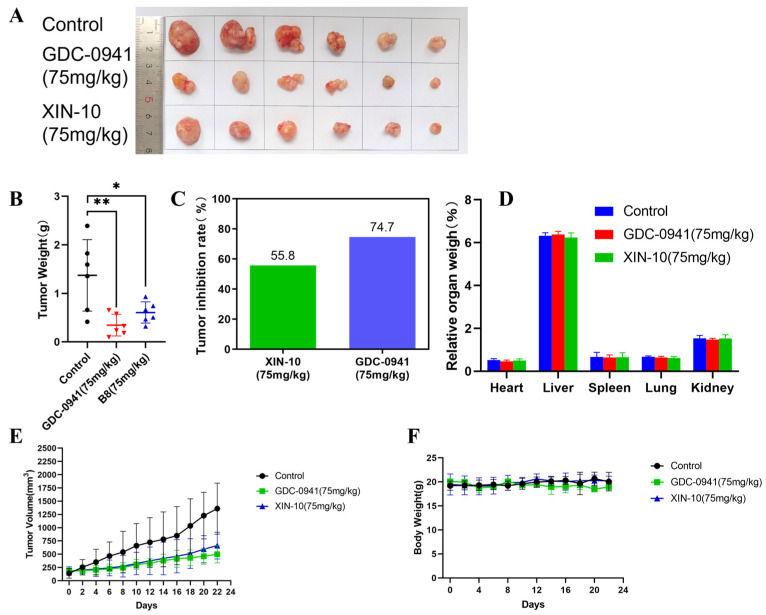
XIN-10 demonstrated strong in vivo efficacy against MCF-7 xenograft tumors in BALB/c nude mice. (**A**) Photographs of excised xenograft specimens obtained during the experiment. (**B**) Mean tumor mass of hairless mice in each group after 21 days of treatment. * *p* < 0.05 indicates a significant difference between the treatment group and the blank control group, while ** *p* < 0.01 indicates a highly significant difference between the treatment group and the blank control group; (**C**) evaluation of tumor suppression rates in each experimental group (TSR, %); (**D**) proportional organ weight of hairless mice in each group after 21 days of treatment; (**E**) mean tumor size in nude mice of each group after 21 days of treatment; and (**F**) mean body mass of hairless mice in each group after 21 days of treatment.

**Figure 15 ijms-24-14821-f015:**
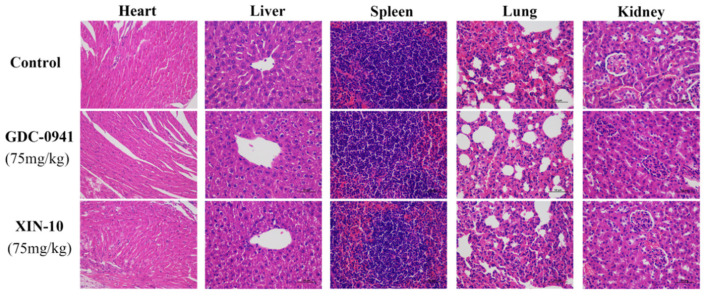
Hematoxylin–eosin staining results of heart, liver, spleen, lung, and kidney tissues of nude mice in each group (Scale bar, 50 µm).

**Figure 16 ijms-24-14821-f016:**
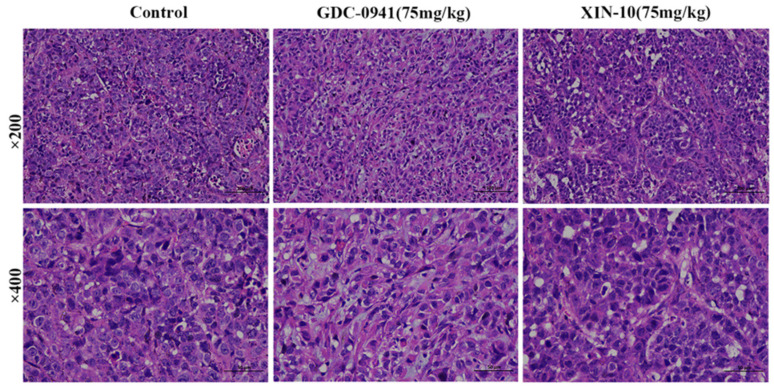
Results of hematoxylin–eosin staining (H&E staining) of MCF-7 transplanted tumors in various groups of nude mice (Scale bar, 50 µm).

**Figure 17 ijms-24-14821-f017:**
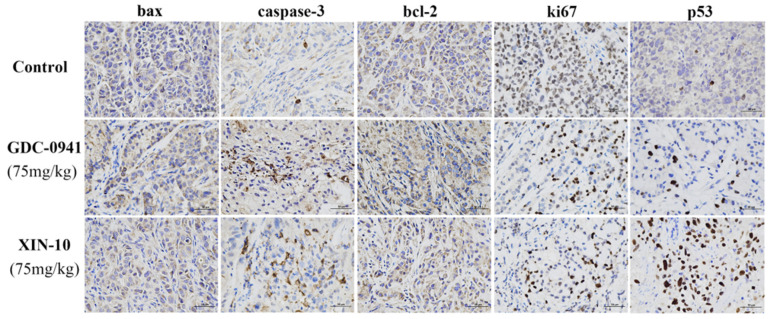
Immunohistochemical staining for bax, caspase-3, bcl-2, ki67, and p65 expression in MCF-7 transplanted tumors (×400, brown is positive expression), Scale bar, 50 µm.

**Figure 18 ijms-24-14821-f018:**
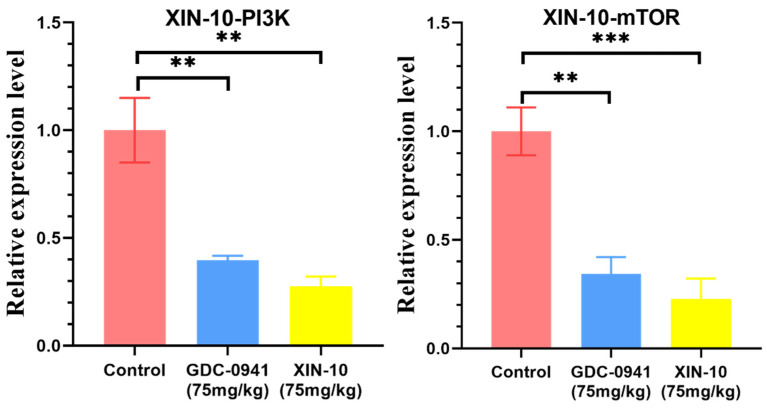
Expression of MCF-7 transplanted tumors to PI3K and mTOR in PCR experiments. ** *p* < 0.01 indicates that XIN10 has a highly significant difference in relative expression level for Control. *** *p* < 0.001 indicates that XIN10 has a very significant difference in relative expression level for Control.

**Table 1 ijms-24-14821-t001:** Cytotoxicity of compounds XIN-10 against 10 cells.

Cancer Cell Lines	IC50 (μM) ^a^	
	XIN-10	GDC-0941
**MCF-7**	**0.211 ± 0.010**	**1.610 ± 0.060**
A549	0.249 ± 0.080	1.263 ± 0.090
Hela	0.861 ± 0.012	3.390 ± 0.160
Hela-MDR	1.702 ± 0.060	0.927 ± 0.010
NCI-H1975	1.510 ± 0.140	0.927 ± 0.070
NCI-H460	0.223 ± 0.060	1.013 ± 0.150
H2228	0.958 ± 0.020	1.855 ± 0.210
Ovcar-3	0.551 ± 0.042	2.125 ± 0.150
U87MG	1.202 ± 0.020	2.214 ± 0.180
MDA-MB-231	7.324 ± 0.560	11.011 ± 0.820

^a^ IC50: concentrations at which cells reduced growth by 50% when treated with the tested compounds for 72 min are expressed as mean IC50 values ± SD of three independent experiments (*n* = 3).

**Table 2 ijms-24-14821-t002:** Inhibition of six kinases by XIN-10.

Compound	Average %Inhibition					
	VEGFR-2	c-Met	mTOR	PI3Kα	AKT1	EGFR^T790M/L858R^
**XIN-10**	5.8 ± 0.4	−4.3 ± 0.7	85.3 ± 2.9	92.9 ± 2.4	−0.7 ± 0.1	6.4 ± 0.4

**Table 3 ijms-24-14821-t003:** The selectivity of XIN-10 towards PI3K and mTOR kinases.

Kinase	Selectivity			
	VEGFR-2	c-Met	AKT1	EGFR^T790M/L858R^
**PI3Kα**	16.1	21.6	132.7	14.5
**mTOR**	14.7	19.8	121.9	13.3

**Table 4 ijms-24-14821-t004:** Enzymatic activities of XIN-10 and GDC-0941 PI3Kα and mTOR.

Compound	IC50 (μM)	
	PI3Kα	mTOR
**XIN-10**	**0.0508**	**0.0214**
**GDC-0941**	**0.0300**	**0.525**

## Data Availability

Not applicable.

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
