# Peer review of "A Novel Dual PI3K/mTOR Inhibitor, XIN-10, for the Treatment of Cancer"

_ijms, 2023, doi:10.3390/ijms241914821_

Round 1
Reviewer 1 Report
The manuscript entitled: A novel dual PI3K/mTOR inhibitor, XIN-10, for the treatment of breast cancer; by Luo L et al., examines the impact of XIN-10 compound on breast cancer. XIN-10 is a compound derived from the modification of GDC-0941 (Pictilisib) which was originally developed by Genentech, and exhibits moderate selectivity (IC50 of 3 nM) for both PI3Kα/δ. The effect on breast cancer was shown in several cell lines, via evaluations of the following parameters: cytotoxic effects, tyrosine kinase selectivity, Acridine Orange DNA staining, FACS for cell cycle and apoptosis analyses, "cloning" (?) colony formation assay, wound migration, qPCR, mitochondrial membrane potential, Western blots for signaling, reactive oxygen species, molecular docking, hemolytic toxicity, and impact on xenograft tumors in vivo. It is demonstrated that XIN-10 is selectively more effective than GDC-0941 with less cytotoxic influence. While it is shown that a significant effect on PI3K/mTOR takes place, yet major flaws in the manuscript are observed.
Major:
1. Cell lines: The title indicates impact on breast cancer. In the abstract it is stated that in-depth were 3 cell lines screened, in the first Result section the impact on 9 cell lines is shown, none (except for one) of which are breast cancer lines. If the goal is to test in breast cancer cell lines MCF7 is the only cell line relevant. It is not accepted to perform evaluations only in one relevant cell line alone. Three breast cancer cell lines should be analyzed. Importantly, evaluations for signaling effect (e.g., impact on p-S6K p, p-AKT and p-4EBP1), xenograft in vivo assay, FACS for cell cycle and apoptosis and perhaps colony formation as well). It is recommended to add MDA-MB-231 breast cancer cell line as also ZR751 and/or BT474.
2. In addition to studying the impact of XIN-10 on PI3K/mTOR by qPCR analyses (it is not clear toward which gene the primers outlined are directed; PI3K: is it toward p110 or p85, for mTOR is it mTORC1 or 2? This should be clearly stated), the direct signaling impact should be evaluated ( e.g., as stated above: p-S6K p, p-AKT and p-4EBP1).
3. The Discussion is insufficient. It merely contains a short paragraph describing (again ) the findings of the manuscript, while it should discuss similar pathway inhibitors in the literature.
Minor points:
- Fig. 2 : should add SD ( statistical analysis).
-Fig. 3: for AO staining, quantification of the apoptotic cells is needed.
-Fig 6B: Colony (and not cloning, please correct!) formation assay is too dense and not acceptable. An adequate dilution for carrying out the assay should be used.
- Fig. 9: What JC-1 stands for? Mitochondrial membrane potential assay (MMP and not MPP?)
- Fig. 11: Signaling for PI3K/mTOR: in the Western blot total AKT is reduced with increased XIN-10 conc. Levels of reduced pAKT should be made as a ratio of pAKT/AKT as also for pmTOR/mTOR.
- Fig. 15. The dose ( e.g., 75 mg/kg) treatment of XIN-10 is huge. What happens at a dose of 5 mg/kg?
- All figures are lacking legends (except for Figs. 12, 14, 15). Fig. 14 follows Fig. 12….
Legends provided are as titles. A descriptive and detailed text should outline each figure.
Remark: On line 31 it is stated: " The PI3K/AKT/mTOR pathway is easily dysregulated…" Easily is not adequate here. Should be: PI3K/AKT/mTOR is a major pathway in cancer.
Author Response
Thanks for your comments on our work. Our manuscript ijms-2566376 was revised according to your comments: The itemized response to each comment was attached.
Major:
- Cell lines: The title indicates impact on breast cancer. In the abstract it is stated that in-depth were 3 cell lines screened, in the first Result section the impact on 9 cell lines is shown, none (except for one) of which are breast cancer lines. If the goal is to test in breast cancer cell lines MCF7 is the only cell line relevant. It is not accepted to perform evaluations only in one relevant cell line alone. Three breast cancer cell lines should be analyzed. Importantly, evaluations for signaling effect (e.g., impact on p-S6K p, p-AKT and p-4EBP1), xenograft in vivo assay, FACS for cell cycle and apoptosis and perhaps colony formation as well). It is recommended to add MDA-MB-231 breast cancer cell line as also ZR751 and/or BT474.
Answer: Thank you very much for your suggestion, the evaluation only in mcf-7 is not reliable to conclude that XIN-10 has an effect on breast cancer, for this reason we added the MDA-MB-231 cell line to carry out MTT experiments, and the results of the experiments showed that the inhibitory effect of XIN-10 in triple-negative breast cancer cells, MDA-MB-231, was not satisfactory, XIN-10 did not have good effects on all breast cancer cell lines, but had good effects on MCF-7-positive breast cancer cell lines, so we changed the title and the abstract accordingly to better express our results. XIN-10 did not inhibit all breast cancer cell lines, but only MCF-7-positive breast cancer cell lines. Therefore, we changed the title and the abstract to better express our results.
- In addition to studying the impact of XIN-10 on PI3K/mTOR by qPCR analyses (it is not clear toward which gene the primers outlined are directed; PI3K: is it toward p110 or p85, for mTOR is it mTORC1 or 2? This should be clearly stated), the direct signaling impact should be evaluated (e.g., as stated above: p-S6K p, p-AKT and p-4EBP1).
Answer: Thank you for pointing out the issue. For the analysis of qPCR, we used primers PIK3CA (p110) and mTORC1 for pathway validation. We have now provided additional explanations for the images and methods, as shown in section 3.7 Quantitative real-time PCR on page 8 and section 4.9 on page 19 Real time PCR section.
- The Discussion is insufficient. It merely contains a short paragraph describing (again) the findings of the manuscript, while it should discuss similar pathway inhibitors in the literature.
Answer: Thank you for pointing out the shortcomings of the article, the discussion section has been expanded and is located in the 3. Discussion section on page 16.
Minor points:
- Fig. 2 : should add SD ( statistical analysis).
Answer: Thank you for your suggestion. SD has been added to Figure 2, located in the 2.2 Evaluation of the kinase inhibitor activities section on page 4.
-Fig. 3: for AO staining, quantification of the apoptotic cells is needed.
Answer: Thank you for your suggestion. AO staining has been counted and quantified into a bar chart representation, located in the 2.3 AO staining section on page 5.
-Fig 6B: Colony (and not cloning, please correct!) formation assay is too dense and not acceptable. An adequate dilution for carrying out the assay should be used.
Answer: Thank you for pointing out the issue. The inappropriate wording has been changed. We apologize for the difficulty in conducting Colony experiments again due to issues with the MCF-7 cell line status and time cycle.
- Fig. 9: What JC-1 stands for? Mitochondrial membrane potential assay (MMP and not MPP?)
Answer: Thank you for your question. JC-1 is an ideal fluorescent probe widely used to detect mitochondrial membrane potential ΔΨm. The transition of JC-1 from red fluorescence to green fluorescence serves as an early detection indicator of apoptosis. And the inappropriate wording has been changed.
- Fig. 15. The dose ( e.g., 75 mg/kg) treatment of XIN-10 is huge. What happens at a dose of 5 mg/kg?
Answer: Due to our mistake, we did not fully explain in the methods section. Our administration method of xenograft tumor nude mice was oral administration instead of intravenous injection. Due to the effects of oral first-pass elimination and other effects, this administration method. The commonly used dosage is much larger than that of intravenous administration. 75 mg/kg is a common dosage of oral administration in nude mouse experiments with xenograft tumors. It has been supplemented in 4.17. Tumor xenograft experiments section.
- All figures are lacking legends (except for Figs. 12, 14, 15). Fig. 14 follows Fig. 12….
Legends provided are as titles. A descriptive and detailed text should outline each figure.
Answer: Thank you for pointing out the problem. The incorrect image has been processed and the image description has been added.
Remark: On line 31 it is stated: " The PI3K/AKT/mTOR pathway is easily dysregulated…" Easily is not adequate here. Should be: PI3K/AKT/mTOR is a major pathway in cancer.
Answer: Thank you very much for your careful check, the inappropriate wording has been changed.

Reviewer 2 Report
This a review of the manuscript by Luo et al. titled “A novel dual PI3K/mTOR inhibitor, XIN-10, for the treatment of breast cancer” submitted to IJMS.
No statistical analyses are presented in the manuscript. Without proper statistical analyses the data presented are not meaningful. Figure 19 is the only figure that indicates a statistically significant difference but no information as to what the significance is or how it was determined is included either in the Methods section or the figure legend. A statistical analyses section must be added to the Methods section and statistical analyses must be included for Figures 1, 2, 4, 5, 6, 7, 11, 14, 15, and 19.
The migration results presented in Figure must be quantified and statistical analyses performed and presented.
The Materials and Methods section must be expanded to include details that would allow other researchers to recreate and repeat the experiments and studies outlined.
Numerous grammatical and spelling errors need to be addressed before a more thorough review can be conducted. The entire manuscript needs to be rewritten.
Numerous grammatical and spelling errors need to be addressed before a more thorough review can be conducted. The meaning is lost due to the poor English. The entire manuscript needs to be rewritten.
Author Response
Thanks for your comments on our work. Our manuscript ijms-2566376 was revised according to your comments: The itemized response to each comment was attached.
No statistical analyses are presented in the manuscript. Without proper statistical analyses the data presented are not meaningful. Figure 19 is the only figure that indicates a statistically significant difference but no information as to what the significance is or how it was determined is included either in the Methods section or the figure legend. A statistical analyses section must be added to the Methods section and statistical analyses must be included for Figures 1, 2, 4, 5, 6, 7, 11, 14, 15, and 19.
Answer: Thank you for pointing out the problem. We performed statistical analysis on the figure and added the statistical analysis method in the Methods section located in 4.18. Statistical analysis section on page 21.
The migration results presented in Figure must be quantified and statistical analyses performed and presented.
Answer: Thanks for your question, migration is now quantified and statistically analyzed in section 3.8 Cell migration on page 9.
The Materials and Methods section must be expanded to include details that would allow other researchers to recreate and repeat the experiments and studies outlined.
Answer: Thanks for your question. Due to our previous mistake, the Materials and Methods section did not appear in the correct place. This section has now been placed in the 4. Methods and Materials section starting on page 17.
Numerous grammatical and spelling errors need to be addressed before a more thorough review can be conducted. The entire manuscript needs to be rewritten.
Answer: Thank you for pointing out the issue. We have made multiple revisions to the article to minimize spelling and grammar errors.

Reviewer 3 Report
The authors described dual PI3K/mTOR inhibitor towards oncology therapeutics. I feel that this manuscript is missing some of important data. When the authors add the data, it may be worth to thinking the publication.
All kinase selectivity and cell based assay of kinase selectivity should be added in the main part.
PK data of the compound during animal model studies should be clarified. In addition, PK/PD studies should be commented.
Author Response
Thanks for your comments on our work. Our manuscript ijms-2566376 was revised according to your comments: The itemized response to each comment was attached.
The authors described dual PI3K/mTOR inhibitor towards oncology therapeutics. I feel that this manuscript is missing some of important data. When the authors add the data, it may be worth to thinking the publication.
All kinase selectivity and cell-based assay of kinase selectivity should be added in the main part.
Answer: Thank you for your suggestion. We have added Tables 3 and 4 to better represent the selectivity of XIN-10 for PI3K and mTOR in section 2.2 Evaluation of the kinase inhibitory activities on page 3.
PK data of the compound during animal model studies should be clarified. In addition, PK/PD studies should be commented.
Answer: Thanks for pointing out the problem. Due to the experimental cycle and consumables, we are sorry that we cannot provide further supplements to PK data and PK/PD studies. The weight changes of the liver, HE staining, and preoperative data from ten items can indirectly respond to metabolic issues.

Reviewer 4 Report
The article, “A novel dual PI3K/mTOR inhibitor, XIN-10, for the treatment of breast cancer” by Luo et al., reports a novel small molecule inhibitor for PI3K/mTOR and its anticancer effects in breast cancer cells and xenograft. The manuscript is relatively well organized and performed. However, the results should be improved as follows:
Major comments
1. A novel small molecule kinase inhibitor may have extra inhibitory activity toward other kinases in human kinome. To add a value for XIN-10, I strongly recommend to check its specificity against human kinome. In addition, IC50 values against PI3K and mTOR and selected kinases in in vitro kinase assays will provide more information for XIN-10.
2. As authors discussed, GDC-0941 is not a good positive control since it failed in phase II clinical studies. To get more clinical relevance, authors are recommended to use a clinical PI3K inhibitor, alpelisib, which has been approved by the US FDA.
3. Since XIN-10 has been developed from GDC-0941, authors have to demonstrate more data for advantage of XIN-10 over GDC-0941. Figure 1 is not sufficient for this.
4. What is the rationale to select cell lines for further studies. In Figure 1, XIN-10 has strongest cytotoxicity in H2228. Why H2228 was not selected?
5. What are the mechanisms for suppressing PI3K and mTOR mRNA expression in Figures 7 and 19?
6. In addition to p-AKT/AKT and p-mTOR/mTOR, other downstream targets such as p-GSK3beta/GSK3beta, eIF4, and p-RPS6/RPS6 are necessary to demonstrate the inhibition of PI3K/mTOR pathway by XIN-10.
7. To declare anticancer effects of XIN-10 in breast cancer cells, authors have to include additional breast cancers.
8. The discussion section is too short. Add more discussion on current status or landscape of PI3K/mTOR inhibitors, advantage of XIN-10 over GDC-0941 and other PI3K/mTOR inhibitors, direction of future developments. Etc/
Minor comments
1. Authors should carefully check typos throughout the manuscript since there are many typos. For example: PI3K/MTOR in line 12, a duplicate of a paragraph in lines 19 – 20, hosphatidylinositol 3-kinase in lines 34 – 35. Authors are also recommended thorough English editing by native speaker(s) to increase readability of the manuscript.
2. The graph in Figure 1 could be improved by pairwise demonstration of XIN-10 and GDC-0941 in the same cells.
3. In the original images, there is no molecular weight markers for each blot.
<The End>
English editing is required.
Author Response
Thanks for your comments on our work. Our manuscript ijms-2566376 was revised according to your comments: The itemized response to each comment was attached.
Major comments
- A novel small molecule kinase inhibitor may have extra inhibitory activity toward other kinases in human kinome. To add a value for XIN-10, I strongly recommend to check its specificity against human kinome. In addition, IC50 values against PI3K and mTOR and selected kinases in in vitro kinase assays will provide more information for XIN-10.
Answer: Thank you for your suggestions. We are sorry that due to consumables and experimental cycle reasons, we are unable to supplement the kinase spectrum. We have added the IC50 values ​​of XIN-10 for PI3K and mTOR kinases, which are located in the 2.2 Evaluation of the kinase inhibitory activities section on page 3.
- As authors discussed, GDC-0941 is not a good positive control since it failed in phase II clinical studies. To get more clinical relevance, authors are recommended to use a clinical PI3K inhibitor, alpelisib, which has been approved by the US FDA.
Answer: Thank you for your suggestions. As we mentioned, the drug GDC-0941 failed in phase II clinical trials, but its anti-proliferative effect and structure still have merits. XIN-10 is modified based on the GDC-0941 structure. The purpose of selecting XIN-10 as a positive control is to explore the advantages of XIN-10 over GDC-041 and to determine whether it can become a potential small molecule inhibitor.
- Since XIN-10 has been developed from GDC-0941, authors have to demonstrate more data for advantage of XIN-10 over GDC-0941. Figure 1 is not sufficient for this.
Answer: Thank you for your question. Regarding this issue, we have added the IC50 part of the kinase, and part 3.7 Quantitative real-time PCR on page 8, and part 3.10 Ros oxygen on page 10, can also jointly reflect the advantages of XIN-10 over GDC-0941
- What is the rationale to select cell lines for further studies. In Figure 1, XIN-10 has strongest cytotoxicity in H2228. Why H2228 was not selected?
Answer: Thank you for your question. Due to a mistake when we filled in the statistical data, the cytotoxicity of XIN-10 against H2228 was actually 0.958μM instead of 0.058μM. We are very sorry for the misunderstanding caused by this mistake. We have now updated the data. and errors on the chart have been corrected.
- What are the mechanisms for suppressing PI3K and mTOR mRNA expression in Figures 7 and 19?
Answer: Thank you for your question. The primers used in these two PCRs are PIK3CA and mTOR1, which have been corrected. The main expression is XIN-10, which reflects the inhibition of the PI3K-AKT-mTOR pathway through the inhibition of these two genes.
- In addition to p-AKT/AKT and p-mTOR/mTOR, other downstream targets such as p-GSK3beta/GSK3beta, eIF4, and p-RPS6/RPS6 are necessary to demonstrate the inhibition of PI3K/mTOR pathway by XIN-10.
Answer: Thank you for your question. Due to the issue of experimental cycle, we are unable to supplement other Western blot experiments. PCR and xenograft tumor experiments in nude mice can also indirectly demonstrate the inhibition of XIN-10 on the PI3K/mTOR pathway.
- To declare anticancer effects of XIN-10 in breast cancer cells, authors have to include additional breast cancers.
Answer: Thank you very much for your suggestions. For this problem, we added MDA-MB-231 cell line to carry out MTT experiment. The experiment results show that the inhibition of XIN-10 in MDA-MB-231, a triple negative breast cancer cell line, is not ideal. XIN-10 does not have a good effect on all breast cancer cell lines, but has a good effect on MCF-7 positive breast cancer cell lines, Therefore, we have made corresponding changes to the title and abstract to better express our experimental results.
- The discussion section is too short. Add more discussion on current status or landscape of PI3K/mTOR inhibitors, advantage of XIN-10 over GDC-0941 and other PI3K/mTOR inhibitors, direction of future developments. Etc/
Answer: Thank you for pointing out the shortcomings of the article, the discussion section has been expanded and is located in the 3. Discussion section on page 16.
